# Use of Waste from Granite Gang Saws to Manufacture Ultra-High Performance Concrete Reinforced with Steel Fibers

**Fernando López Gayarre** *, **Jesús Suárez González**, **Iñigo Lopez Boadella**, **Carlos López-Colina Pérez** and **Miguel Serrano López**

Polytechnic School of Engineering—University of Oviedo, Campus de Viesques, 33203 Gijón, Spain; suarezg@uniovi.es (J.S.G.); inigo2208@hotmail.com (I.L.B.); lopezpcarlos@uniovi.es (C.L.-C.P.); serrano@uniovi.es (M.S.L.)

* Correspondence: gayarre@uniovi.es; Tel.: +34-985182278

**Abstract:** The purpose of this study is to analyze the feasibility of using the ultra-fine waste coming from the granite cutting waste gang saws (GCW-GS) to manufacture ultra-high performance, steel-fiber reinforced concrete (UHPFRC). These machines cut granite blocks by abrasion using a steel blade and slurry containing fine steel grit. The waste generated by gang saws (GCW-GS) contains up to 15% $Fe_2O_3$ and up to 5% $CaO$. This is the main difference from the waste produced by diamond saws (GCW-D). Although this waste is available in large quantities, there are very few studies focused on recycling it to manufacture any kind of concrete. In this study, the replaced material was the micronized quartz powder of natural origin used in the manufacture of UHPRFC. The properties tested include workability, density, compressive strength, elasticity modulus, flexural strength, and tensile strength. The final conclusion is that this waste can be used to manufacture UHPFRC with a better performance than that from diamond saws given that there is an improvement of their mechanical properties up to a replacement of 35%. Even for higher percentages, the mechanical properties are within values close to those of control concrete with small decreases.

**Keywords:** ultra-high performance concrete; waste; granite gang saws; steel fibers; compressive strength; flexural strength; elasticity modulus



## 1. Introduction

The construction sector is one of the industrial activities that consume the most natural resources and generate the most waste in the European Union. Most of the waste generated is mineral waste, i.e., 2/3 of the total according to the waste statistics [1] of the 2018 Eurostat. For this reason, many studies are being carried out regarding the manufacture of concrete using recycled materials or by-products from the construction sector as a partial or total replacement of natural aggregates. Among the activities that consume the most natural resources and generate the greatest amount of waste is the extractive industry. These wastes can be extremely difficult and expensive to recycle. According to the 2018 Eurostat regarding waste in the European Union, the mining and quarrying sector generates 26% of the total waste, and it was the second highest generator of waste in the EU in 2018 [1].

In recent years, various studies have analyzed the use of mining waste in the manufacture of concrete. Several of these works are aimed at replacing some of the components of ultra-high performance fiber reinforced concrete (UHPFRC), with different types of waste. N. A. Soliman and A. Tagnit-Hamou demonstrated in several studies [2–4] that glass waste could be included in ultra-high performance concrete (UHPC). They found that it was possible to replace 50% of the quartz sand with glass waste without affecting the compressive strength of UHPC [2]. They also showed that 20% of the cement and up to 100% of the micronized quartz could be replaced, obtaining concrete with a compressive strength of 220 MPa, a flexural strength of 29 MPa, and a modulus of elasticity of 55 GPa [4]. These good results can be attributed to the pozzolanicity of powder glass, its high strength,

and its elastic modulus. However, it should be mentioned that for a 100% of substitution with glass sand, they have found an increase of the porosity of 21%. The increase of the porosity has also been reported by Zhao et al. [5], who observed a rise over 90% with respect to the control mix when they substituted 100% of fine coarse with iron ore tailings as fine aggregate. Another study [3] of N. A. Soliman and A. Tagnit-Hamou showed that using powder glass as an alternative to silica fume improved the compressive strength of UHPC in a 15% for a substitution of 30% [3]. Altogether, their results prove the viability of an economical UHPC with good mechanical properties and a reduced carbon footprint. Zhigang Zhu et al. [6] investigated the influence of iron ore tailings (IOT) as an alternative to fine aggregate on the properties of UHPC. The results showed a higher compressive strength of UHPC when using IOT with a maximum size of 1.18 mm as a replacement for quartz sand. The best performance was achieved for a replacement of 60%. Sujing Zhao et al. also analyzed the influence of this waste as an alternative to natural aggregate [5]. Their results showed a loss of strength in the UHPC when incorporating the IOT. However, they found that the addition of 2% steel fibers to UHPC with 20–40% IOT improved the mechanical properties, making them comparable to those of the reference concrete. Zhao et al. attributed this improvement to a better dispersion of the fibers caused by the IOT, making a more homogeneous matrix. Other researchers have studied the possibility of incorporating other wastes, such as copper slag. The results obtained by S. Al-Jabri et al. [7] showed an increase in the compressive, flexural, and tensile strength of HPC when using copper slag as an alternative to fine aggregate, while always maintaining the same workability of the mixture. For a 100% replacement, there are increases in compressive strength above 19%. Similarly, the results obtained by PS Ambily et al. [8] demonstrated the viability of manufacturing UHPC with a compressive strength greater than 150 MPa when incorporating copper slag.

Granite cutting wastes (GCWs) have also been used to substitute some of the components of mortars or concretes. O. Mashaly et al. [9] studied the feasibility of incorporating granite sludge as an alternative to cement in the manufacture of concrete and mortar. The results showed that for a 20% substitution, the variations in the mechanical properties are not relevant, as they observed an increase in resistance to abrasion and freeze-thaw cycles. For their part, S. Singh et al. [10,11] analyzed incorporating granite cutting waste as an alternative to fine aggregate in order to obtain more sustainable concretes. The results showed that using 25–40% cutting waste granite has a positive effect on the strength and durability of concrete. F. Kala [12] obtained similar results when incorporating granite powder as an alternative to fine aggregate in the manufacture of high-performance concretes. Most of the studies carried out focus on the use of waste from the extractive industry to make conventional concrete or HPC, while few of them focus on the use of this waste for the manufacture of UHPC or UHPFRC. J. Suárez et al. [13] analyzed the influence of incorporating waste from fluorite mines as an alternative to fine aggregate in the manufacture of UHPFRC. The results show that a 70% substitution provides acceptable values of compressive strength, flexural strength, and tensile strength. The results obtained by I. López et al. [14] are also along this line, and in their study they substituted micronized quartz for granite cutting waste in UHPC. The properties obtained for substitutions of 70% are comparable to those of the control concrete, while, for a ratio of 35%, an increase in compression, flexural, and tensile strength of the concrete was observed. In a previous work carried out by I. López et al. [14], the substitution of micronized quartz by fine granite waste from diamond saws cutting machines (GCW-D) was analyzed with favorable results. The results showed an improvement in workability and compressive strength for a total replacement of micronized quartz. Both flexural strength and tensile strength increased for a 35% substitution, while for a replacement of 70%, the values obtained showed little variation with respect to the control concrete.

In the present study, granite cutting waste from granite cutting machines (GCW-GS) called gang saws are used as an alternative to micronized quartz. Gang saws are a very economical system used as an alternative to diamond saws to cut granite blocks. With this

type of machine, the granite block is cut by the abrasive effect of the forward and backward movements of steel plates watered by an aqueous mixture containing steel particles kept in suspension by the densification of the mixture with calcium oxide (CaO) (Figure 1). These wastes are different from those used in [14] as the waste is mixed with up to 15% $Fe_2O_3$ and up to 5% CaO from the aqueous grout used in the cutting process of gang saws. The main objective of this work is to evaluate the behavior of UHPFRC when this waste is used. In addition, this work contrasts these results with those obtained in the previous study in which wastes free of $Fe_2O_3$ and CaO were used, in order to analyze the influence of these two substances on the mechanical properties of UHPFRC.

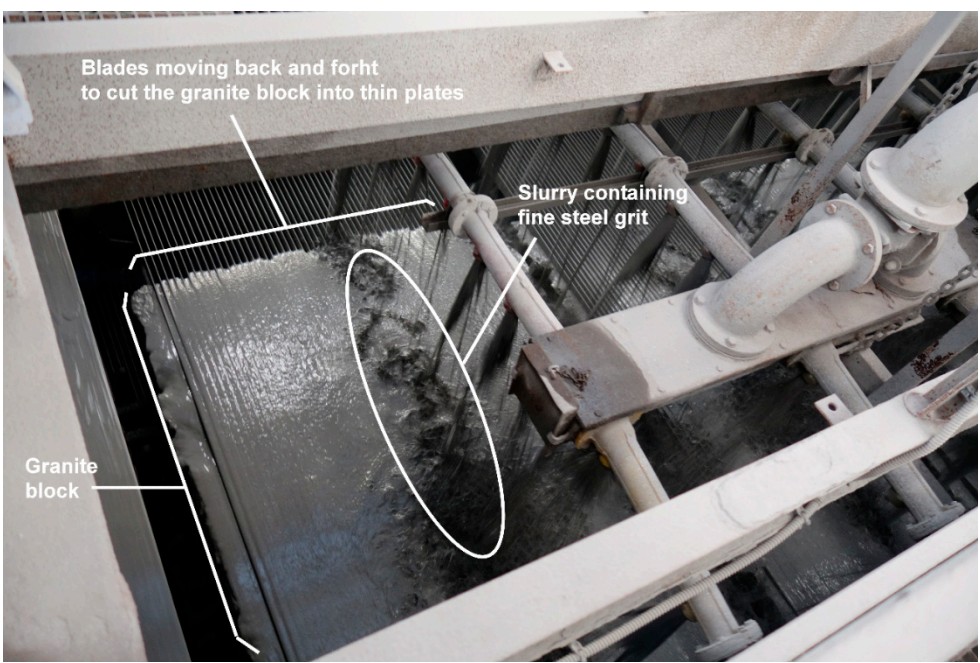

**Figure 1.** Granite gang saw cutting machine.

## 2. Experimental Study

### 2.1. Materials

The cement used was CEM I 42.5 R/SR, supplied by Lafarge-Holcim S.A. (Madrid, Spain). The properties of the cement meet the requirements of the EN 197-1 standard [15] and comply with the recommendations of the EHE-08 [16]. For natural aggregates, two fractions of silica sand were used. The sands, with two granulometric fractions 0/0.5 mm and 0.5/1.6 mm, were supplied by Sílices La Cuesta (Asturias, Spain). For the additions, densified silica fume (Elkem Microsilica® 940) with a mean particle size of 0.15 μm and micronized quartz with a maximum particle size of 40 μm, supplied by Silicas Gilarranz S.A (Segovia, Spain), were used. The short steel fibers, with a diameter of 0.2 mm and a length of 13 mm, were supplied by Arcelor Mittal (Asturias, Spain). To achieve a workable UHPFRC, a polycarboxylate superplasticizer, provided by ViscoCrete-225 Powder (Madrid, Spain) was used. As an alternative to micronized quartz, granite cutting waste from gang saw cutting machines (GCW-GS) was used (Figure 1).

Table 1 shows the values of the particle density, water absorption, and humidity of the materials used in this study. The particle density was determined by means of a helium pycnometer, while the specifications of the EN 1097-6 standard were followed to evaluate the water absorption of silica sands [17]. The higher density of GCW-GS, in relation to the additives used in the control concrete and GCW-D, is due to the presence of iron oxide particles.

**Table 1.** Density, water absorption, and humidity of materials.

| Property | Sand (0.5/1.6 mm) | Sand (0/0.5 mm) | Silica Fume | Micronized Quartz | GCW-GS |
|---|---|---|---|---|---|
| Bulk density (kg/m³) | 2616 | 2616 | 2300 | 2609 | 2856 |
| Absorption at 24 h | 0.53% | 0.28% | - | - | - |
| Humidity (%) | 0% | 0% | <3.00% | <0.2% | 0% |

Table 2 shows the chemical composition of the waste (GCW-GS), determined by X-ray fluorescence. The differences between this GCW-GS waste and that of the GCW-D used in the previous article [14] are the higher content of $Fe_2O_3$ (14.59%), due to the presence of steel particles, and the 4.53% CaO. The higher percentage of $SiO_2$ is attributable to the difference in the type of granite that was cut to generate the two wastes.

**Table 2.** Chemical analysis (%) of the granites powder and micronized quartz.

| Specimen | $SiO_2$ | $Al_2O_3$ | $Fe_2O_3$ | MnO | MgO | CaO | $Na_2O$ | $K_2O$ | $TiO_2$ | $P_2O_5$ | L.O.I |
|---|---|---|---|---|---|---|---|---|---|---|---|
| Micronized quartz | >99.3 | 0.26 | 0.05 | - | - | 0.02 | - | 0.04 | 0.05 | - | - |
| GCW-GS | 60.51 | 11.50 | 14.59 | 0.13 | 0.41 | 4.53 | 2.68 | 4.08 | 0.24 | 0.15 | 0.73 |

Figure 2 shows the granulometric distribution of the different materials used in this study. It can be seen that granite waste have a particle size close to micronized quartz, and therefore it is proposed as an alternative to replace this material. Although silica fume has a maximum particle size of 0.15 μm, the particle size distribution observed in Figure 2 is due to the fact that it is densified.

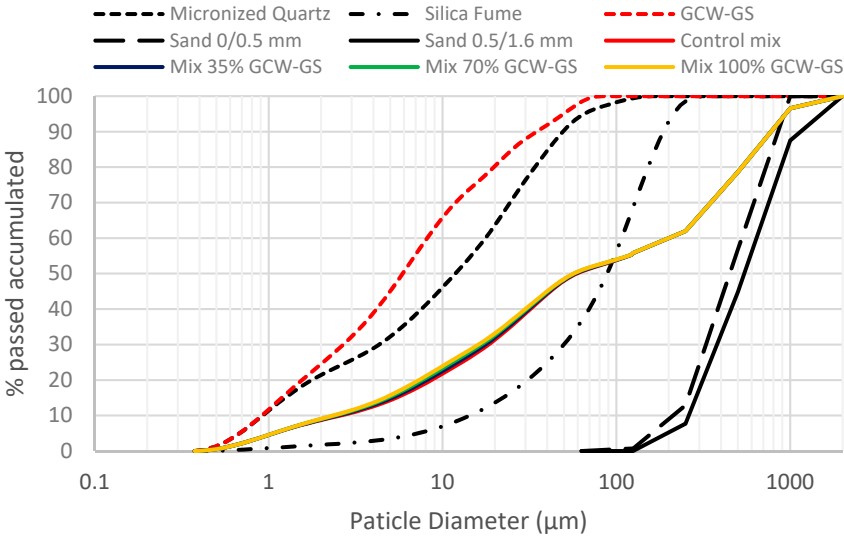

**Figure 2.** Granulometric curves of materials and resulting mixes.

The granulometric curve resulting from the combination of all the components of the concrete mixtures with the different substitutions of GCW-GS is also shown in Figure 2. As the percentage of substitution rises, there is a slight improvement in the granulometric distribution, following the criteria of ideal packing curves proposed by authors such as Andreasen as well as Funk and Dinger [18,19]. It therefore follows that the influence of the granulometry on the final result will lead to a slight improvement in the packing.

### 2.2. Mix Design

To carry out this study, the mix proportions of the UHPFRC were controlled until we achieved a self-compacting control UHPFRC with a compressive strength above 110 MPa. Once the control dosage was established, the experimental program continued, substituting 35%, 70%, and 100% of the micronized quartz for the same volume of granite cutting waste

(GCW-GS). Table 3 shows the dosages of the resulting UHPFRCs. The higher the percentage of GCW-GS substitution, the greater the amount of superplasticizer additive (SP) needed. This is due to the reaction of the water with the CaO present in this waste, which dehydrates the mixture.

**Table 3.** Mix proportions of ultra-high performance steel fiber reinforced concrete (UHPFRC) (kg/m$^3$).

| Specimen | Cement | Sand (0/0.5) | Sand (0.5/1.6) | Micronized Quartz | Granite Sludge Powder | Silica Fume | Water | SP | Steel Fibers |
|---|---|---|---|---|---|---|---|---|---|
| Control | 800 | 302 | 565 | 225 | - | 175 | 175 | 10 | 160 |
| 35% GCW-GS | 800 | 302 | 565 | 146 | 79 | 175 | 175 | 15 | 160 |
| 70% GCW-GS | 800 | 302 | 565 | 68 | 158 | 175 | 175 | 16 | 160 |
| 100% GCW-GS | 800 | 302 | 565 | - | 225 | 175 | 175 | 18 | 160 |

*2.3. Experimental Program*

The experimental program includes a total of four mixes. The first mix is the control UHPFRC, and the other three are UHPFRC with GCW-GS in different percentages.

The mixing procedure was as follows. First, the silica sands and the micronized quartz or the corresponding granite waste were put in the mixer. Next, the silica fume and cement were added. The materials were mixed for 30 s before including the water. After 2 min 30 s from the beginning of the mixing, the superplasticizer and the steel fibers were added. The mixing process ended after 25 min.

The specimens were manufactured according to the specifications of the EN 12390-1 standard [20]. A total of nine specimens were manufactured per mix: three prismatic specimens of 10 × 10 × 40 cm, three cubic specimens of 10 × 10 × 10 cm, and three cylindrical specimens of 15 × 30 cm. The specimens were removed from the molds after 24 h and transferred to a humid chamber where they remained for 28 days at a temperature of 20 °C and a relative humidity of 95% (Figure 3), as specified in the EN 12390-2 standard [21].

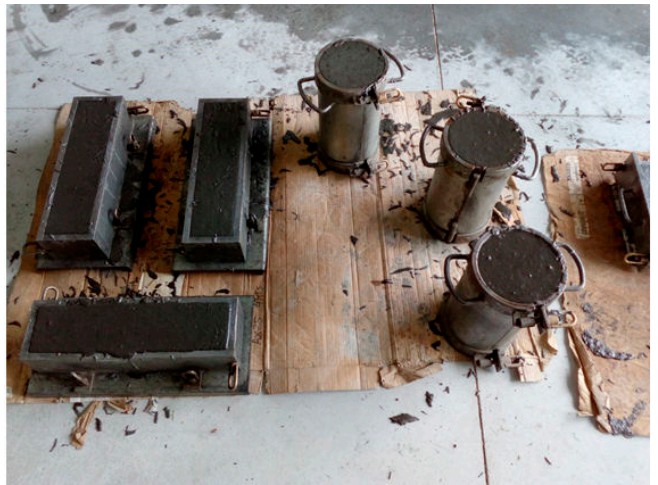
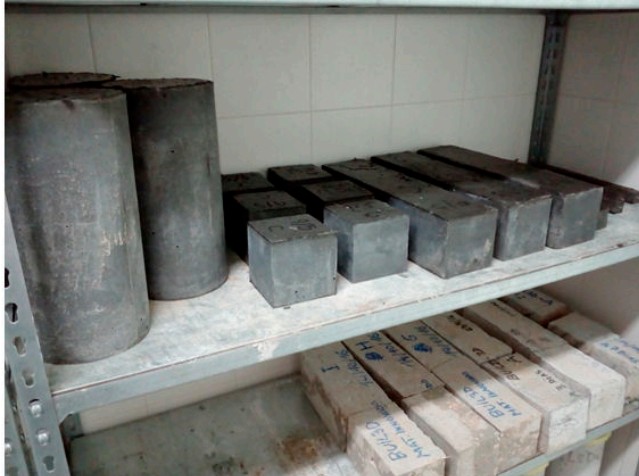

**Figure 3.** Preparation and drying of samples.

The following properties were evaluated: the consistency of the fresh mix, density of hardened concrete, compressive strength, modulus of elasticity, flexural strength, and tensile strength.

Physical Properties

The consistency of the fresh UHPFRC was determined once the mixing process was finished and performed according to the NF P 18-470 standard [22]. The density of the hardened UHPFRC was calculated following the specifications of the EN 12390-7 standard [23]. This was done using the three 15 × 30 cm cylindrical specimens cast per mix. The

compressive strength was determined using the three cubic specimens of $10 \times 10 \times 10$ cm, following the indications of the EN 12390-3 standard [24]. The modulus of elasticity was determined according to the EN 12390-13 standard [25] using the three $15 \times 30$ cm cylindrical specimens. The flexural strength of the UHPFRC were calculated according to the NF P 18-470 standard [22] on the three prismatic specimens of $10 \times 10 \times 40$ cm (Figure 4). To determine tensile strength, a new method based on impact loading was proposed in [26], but it is not applicable to this study due to the using of fibers; instead, this property was determined by an inverse analysis from the stress–strain curves obtained from the flexural strength test, following the methodology proposed by Lopez Martínez in his doctoral thesis [27]. Table 4 shows the mean value of the results obtained from testing three specimens for each percentage of substitution.

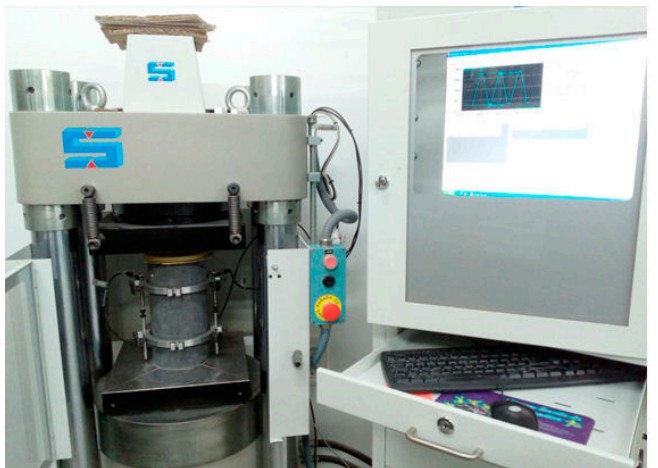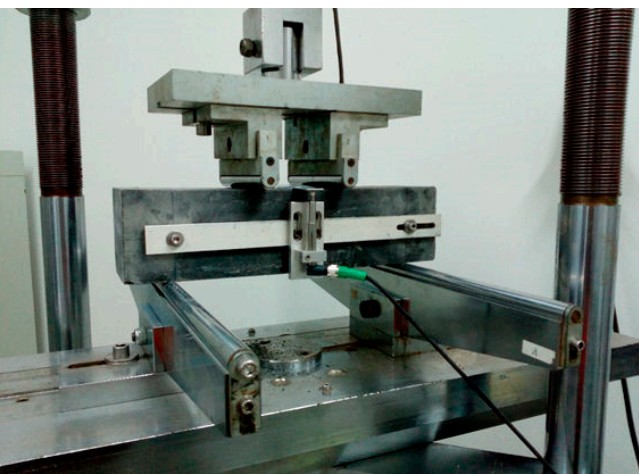

**Figure 4.** Modulus of elasticity and flexural strength tests.

**Table 4.** Average results of UHPFRC.

| Properties | Control | | 35% GCW-GS | | 70% GCW-GS | | 100% GCW-GS | |
|---|---|---|---|---|---|---|---|---|
| | Value | Variation | Value | Variation | Value | Variation | Value | Variation |
| Slump (cm) | 25 | | 23 | | 20 | | 21 | |
| Density (kg/m$^3$) | 2410 | $\pm$35 | 2420 | $\pm$44 | 2470 | $\pm$35 | 2500 | $\pm$14 |
| Compressive strength (MPa) | 117 | $\pm$2.9 | 127 | $\pm$6.5 | 134 | $\pm$4.6 | 129 | $\pm$6.5 |
| Modulus of elasticity (GPa) | 45 | $\pm$1.0 | 43.4 | $\pm$1.2 | 42 | $\pm$1.6 | 41 | $\pm$0.1 |
| Flexural strength (MPa) | 23 | $\pm$0.8 | 25.7 | $\pm$0.3 | 19 | $\pm$1.6 | 22 | $\pm$3.4 |
| Tensile strength (MPa) | 8.7 | $\pm$0.6 | 11.8 | $\pm$0.1 | 6.8 | $\pm$1.2 | 8.8 | $\pm$0.8 |

## 3. Analysis of Results

As mentioned above, the particularity of the granite wastes from gang saws used in this test (GCW-GS) is that they contain higher percentages of CaO (4.5%) and $Fe_2O_3$ (14.6%) than the granite cutting waste from diamond saws (GCW-D) [14]. Therefore, the results obtained will be related, in some way, to the presence of these compounds in the manufactured concrete. Both CaO and $Fe_2O_3$ are occasionally used as additives in the form of nanoparticles to change the properties of mortars and concretes. For this reason, to facilitate the analysis of results, the percentage increase of $Fe_2O_3$ and CaO with respect to the amount of cement used in the different mixtures was calculated (Table 5). The use of GCW-GS in substitutions of 35%, 70%, and 100% is equivalent to the addition of 1.43%, 2.88%, and 4.1% of $Fe_2O_3$, and of 0.44%, 0.88%, and 1.26% of CaO, respectively, in relation to the amount of cement in the mix.

**Table 5.** Percentage of Fe₂O₃ and CaO relative to the amount of cement in the mix.

| Specimen | Cement (kg/m³) | GCW-GS (kg/m³) | Fe₂O₃ (14.6% GCW-GS) (kg/m³) | CaO (4.5% GCW-GS) (kg/m³) | % Fe₂O₃ over Cement | % CaO over Cement |
|---|---|---|---|---|---|---|
| Control | 800 | - | - | - | - | - |
| 35% GCW-GS | 800 | 79 | 11.5 | 3.55 | 1.43% | 0.44% |
| 70% GCW-GS | 800 | 158 | 23.06 | 7.11 | 2.88% | 0.88% |
| 100% GCW-GS | 800 | 225 | 32.85 | 10.12 | 4.1% | 1.26% |

Iron oxide ($Fe_2O_3$) is often used as an additive in mortar and concrete in percentages of up to 5% of the amount of cement because the pozzolanic reactions it causes improve the mechanical strength [28–31].

Calcium oxide (CaO) is also used as an additive in mortars and concretes in percentages of up to 5% to reduce shrinkage without adverse effects on the compressive strength in high-performance concrete. This is due to its pozzolanic effect when used in combination with silica fume [32].

### 3.1. Workability

The results obtained for consistency are shown graphically in Figure 5 together with those obtained in the previous article [14] for GCW-D wastes. Very different behaviors were observed with the two different granite wastes. The consistency of fresh UHPFRC increased when GCW-D was used as an alternative to micronized quartz, while GCW-GS produced a loss in the consistency. The inferior workability of the mix with GCW-GS is most likely due to the presence of CaO in the mix [33], which reacts with water to produce calcium hydroxide and thereby interfering with the cement hydration process. Although to some degree this is compensated for by adding a superplasticizer, its effect is still evident.

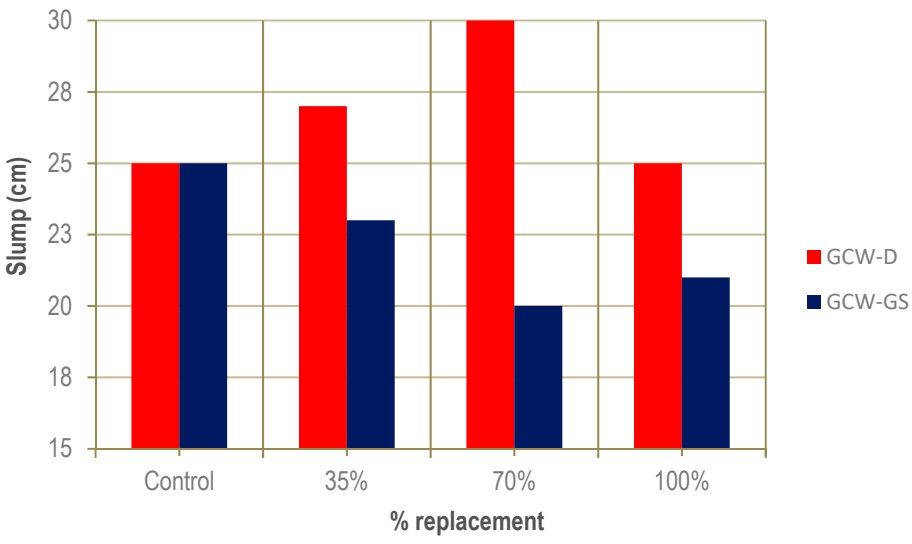

**Figure 5.** Slump of UHPFRC for both wastes (GCW-D [14] and GCW-GS).

The loss of workability for UHPFRC with GCW-GS when increasing the percentage of substitution concurs with the results obtained in other studies when incorporating the waste of granite stone slurry as an alternative to the finest aggregate [34].

### 3.2. Density

Figure 6 shows the average results obtained from the density tests on the different mixes of hardened UHPFRC. The results show a higher density for the mixes with GCW-GS and an increase in density as the percentage of substitution increases. This is due to the higher particle density of GCW-GS, i.e., 2856 kg/m³, compared to micronized quartz, i.e., 2609 kg/m³. In the case of UHPFRC with GCW-D, although a reduction in density was

observed, the variation is less than 2% and is within the error bars (Figure 6). In summary, the incorporation of GCW-GS produced an increase in the density of the concrete while the incorporation of the residue GCW-D did not affect this property.

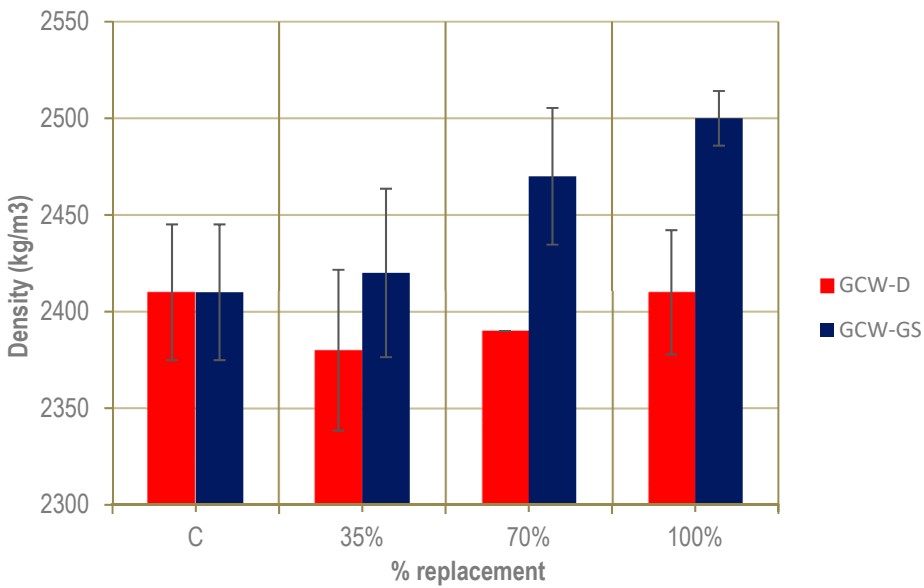

**Figure 6.** Density of UHPFRC for both wastes (GCW-D [14] and GCW-GS).

### 3.3. Compressive Strength

Figure 7 shows the results obtained. They correspond to the mean values obtained in the tests for each mix. Firstly, an increase in compressive strength can be seen for the different substitution percentages, regardless of the type of waste. The results obtained were above 120 MPa, reaching a compressive strength of 133.8 MPa for 70% GCW-GS. The strength of concrete with GCW-GS was around 5% better than concrete made with GCW-D. There are two factors that can explain this improvement. Firstly, there was a slight improvement in packing produced by GCW-GS, as can be seen in the granulometric curves in Figure 2. Secondly, the presence of $Fe_2O_3$ and CaO in the mixture improves pozzolanic activity [28–31] and reduces autogenous shrinkage [32]. The favorable effect of $Fe_2O_3$ occurs for substitution percentages of around 2–3% on the total quantity of cement [30,35]. This explains the slight decrease in compressive strength for the 100% substitution with GCW-GS (equivalent to 4% of $Fe_2O_3$).

These increases in the compressive strength are agreement with the results obtained in other studies where the influence of the use of granite waste in different types of concrete has been analyzed [10,12,14]. Dr. Kala [12] used granite fines as an alternative to fine aggregates in the manufacture of high performance concrete. He obtained increases in the compressive strength between 6.12% and 22.14% depending on the percentage of substitution. S. Singh [10] observed an increase in the compressive strength of concrete with a replace up to 30% of the fine aggregate by granite cutting waste. He attributes this improvement to the formation of a dense and compact matrix because the granite particles are fine enough to fill a large part of the voids present in the concrete. I. López et al. [14], also attribute the improvement in their results obtained in compressive strength from UHPC, by substituting micronized quartz for granite residues, to the greater number of fine particles present in these type of waste.

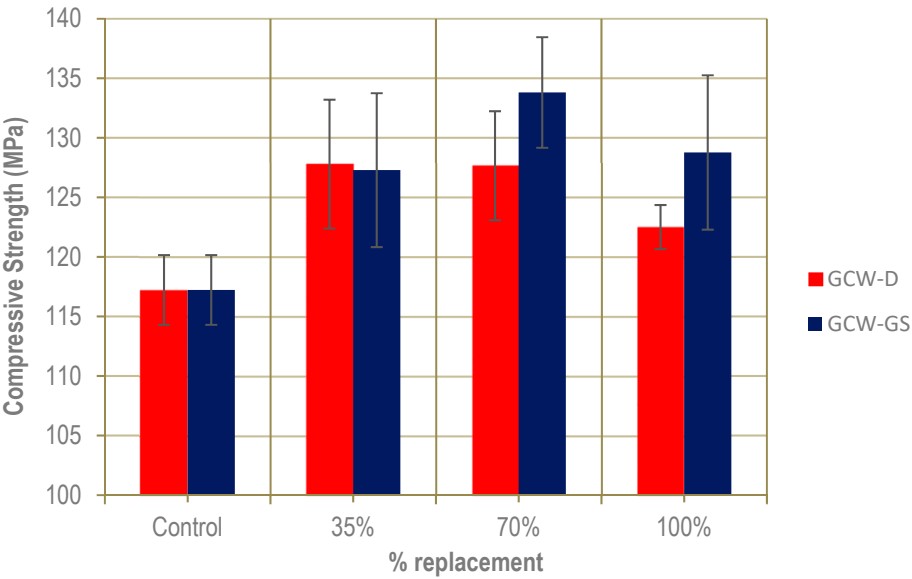

**Figure 7.** Compressive strength of UHPFRC for both wastes (GCW-D [14] and GCW-GS).

### 3.4. Elasticity Modulus

Figure 8 shows the mean values of the elasticity modulus for the different mixes as a function of the percentage of substitution and the type of waste (GCW-D or GCW-GS). The results for the GCW-GS show a slight decrease in the modulus of elasticity with respect to the control concrete. For concrete with GCW-D, the variations are negligible, i.e., 3% for a substitution of 100% GCW-D. In the case of GCW-GS, a greater reduction in the modulus of elasticity was observed, i.e., between 3.5% and 8.5%, depending on the percentage of substitution.

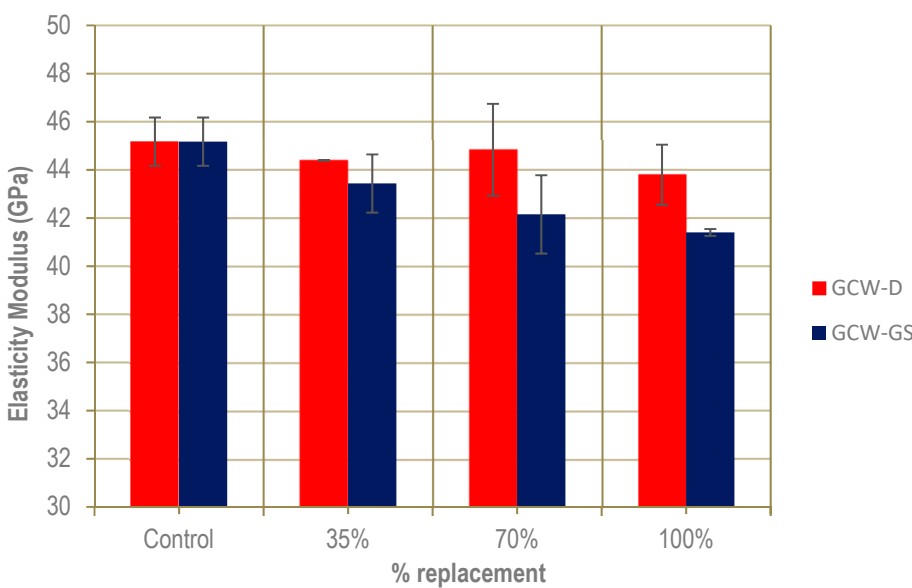

**Figure 8.** Elasticity modulus of UHPFRC for both wastes (GCW-D [14] and GCW-GS).

Overall, the modulus of elasticity tends to decrease with most of the substitute materials used in the studies discussed below. Pyo and H. K. Kim [36] did not observe significant variations in the modulus of elasticity of concrete when incorporating different types of ash as an alternative to silica powder. For a ratio of 25% fly ash, the variation was less than 6.5%. J. Suárez et al. [13] also observed a slight variation when incorporating waste mining sand (WMS) as an alternative to finest silica sand. For a replacement of 100% WMS, the loss was less than 6%. However, in other studies, the results showed more significant

variations. Yacizi et al. [37] analyzed the influence of using blast furnace slag and fly ash on the properties of reactive powder concrete. A decrease in the modulus of elasticity of 10% and 18% were observed when 30% and 40% of blast furnace slag, respectively, was incorporated. However, Safiuddin et al. [38] observed an increase of between 2% and 3% in the modulus of elasticity of high-performance concrete when using industrial by-products. This was related to a reduction in its porosity.

### 3.5. Flexural Strength

Figure 9 shows the effects of granite wastes on the flexural strength of UHPFRC. The high variability in the results may be due to the fact that the steel fibers are not evenly distributed in the matrix. For a substitution of micronized quartz of 35%, a slight increase in flexural strength was observed over the control concrete with both types of wastes. For 35% GCW-D, this increase was 6%, while for 35% GCW-GS it was 12%. However, for higher percentages of substitution, a decreasing trend in flexural strength was observed. As with compressive strength, the presence of $Fe_2O_3$ in percentages greater than 2–3% of the total cement (equivalent to a 70–100% substitution with GCW-GS) was found to be the cause of the loss of strength [30,35]. In any case, all the results were in a range of values close to those of the control concrete.

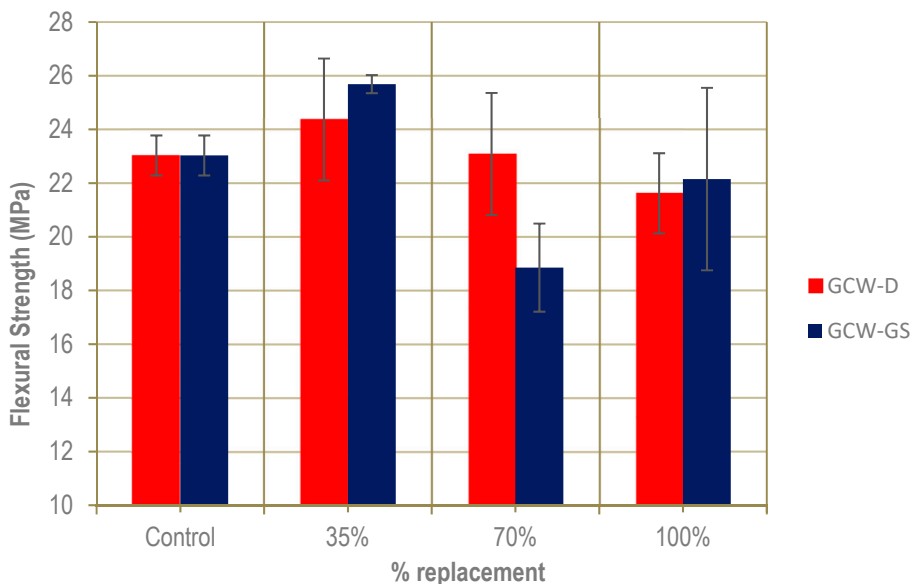

**Figure 9.** Flexural strength of UHPFRC for both wastes (GCW-D [14] and GCW-GS).

These results are similar to those obtained in other studies, where the influence of using granite waste in the manufacture of different concretes was analyzed. The results obtained by S. Singh [10] showed an increase in flexural strength when replacing up to 50% of the fine aggregate in conventional concrete. S. Singh attributed this improvement to a better bond with the cement paste due to the irregular shape of granite. Similar results were obtained by I. Lopez et al. [14] by using granite cutting waste as an alternative to micronized quartz in UHPC. The results showed an increase in flexural strength for 35%, with respect to the reference UHPC.

### 3.6. Tensile Strength

To determine the tensile strength, an indirect method was followed. We took the stress–strain curves obtained in the flexural test as the starting point, and we followed the methodology proposed by López Martínez in his doctoral thesis [27]. First of all, three key points have to be determined, according to Figure 10.

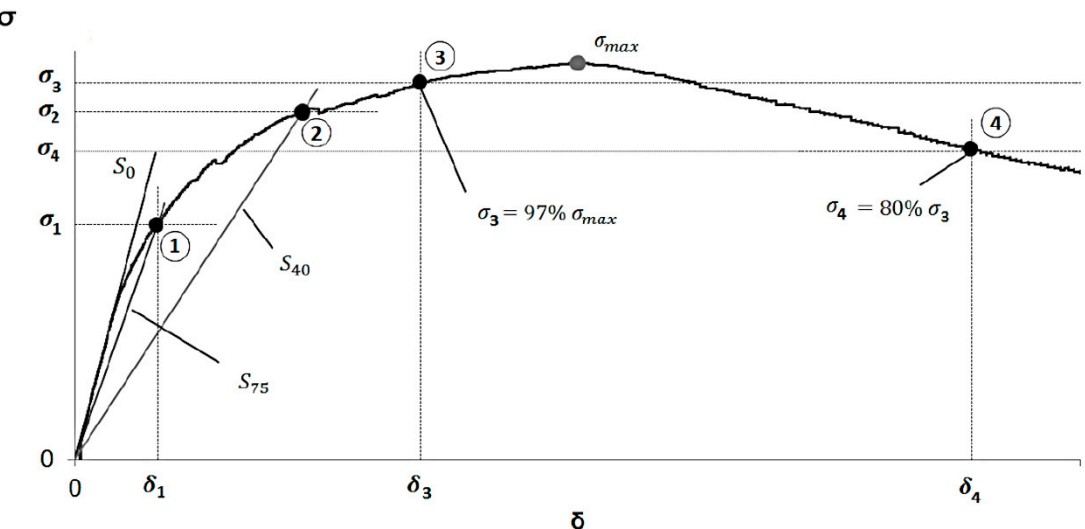

**Figure 10.** Stress–strain curves and key points.

- Point 1 is the intersection of the curve test with a line of a slope equal to 0.75 of the slope of the elastic zone of the curve.
- Point 2 is the intersection of the curve test with a line of a slope equal to 0.40 of the slope of the elastic zone of the curve.
- Point 3 is the point of the ascending zone of the curve, with 97% of the highest stress.

Once we obtained these points, the tensile strength of the UHPFRC can be determined using the following equations [27].

$$E = 2.40 \, h \, m$$

$$f_t = \frac{\sigma_{75}}{1.63} \left( \frac{\sigma_{75}}{\sigma_{40}} \right)^{0.19}$$

$$\varepsilon_{t,u} = \frac{f_t}{E} \left( 7.65 \frac{\delta_{loc}}{\delta_{75}} - 10.53 \right)$$

$$\varepsilon_{t,el} = f_t / E$$

$$\alpha = \varepsilon_{t,u} / \varepsilon_{t,el}$$

$$f_{t,u} = \alpha^{-0.18} \left( 2.46 \frac{\sigma_{loc}}{\sigma_{75}} - 1.76 \right) f_t$$

In these equations, $E$ represents the elasticity modulus, $m$ represents the slope of the elastic region of the stress–strain curve, $h$ is the thickness of the specimen in mm, $f_t$ is the crack resistance of the matrix reinforced with fibers, $f_{t,u}$ is the ultimate tensile strength, and $\varepsilon_{t,u}$ is the peak deformation.

Figure 11 shows the mean values obtained for the tensile strength of UHPFRC. The results obtained replicate those obtained in the preceding test. For a ratio of 35%, an increase in tensile strength was observed, reaching 10.8 MPa for GCW-D and 11.8 MPa for GCW-GS. These are increases of 24% for GCW-D and 35% for GCW-GS. However, for substitutions of 70% or more, the values obtained are similar to those of the control UHPFRC. Except for the 70% GCW-GS where a loss of 22% was observed, the variations are less than 5% and are within the error bars of the test.

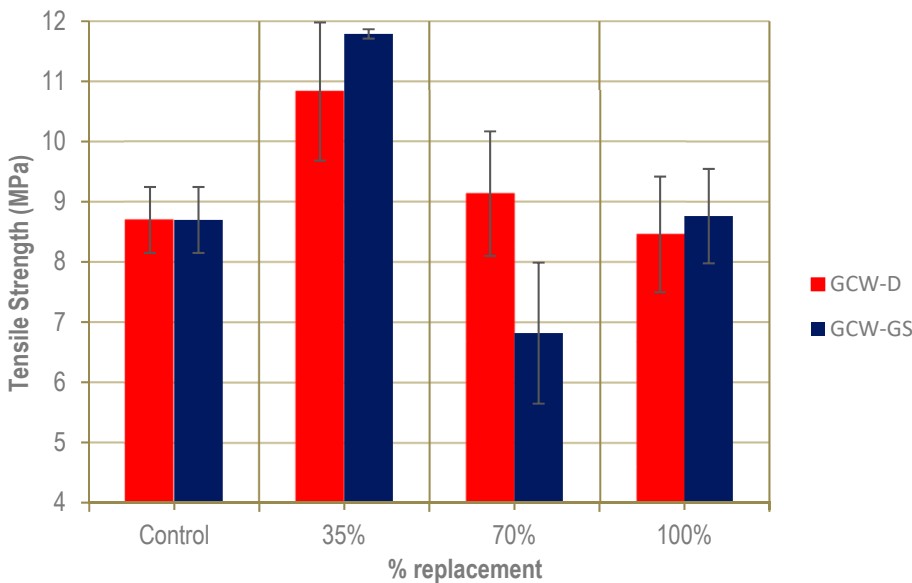

**Figure 11.** Tensile strength of UHPFRC for both wastes (GCW-D [14] and GCW-GS).

The presence of iron oxide explains both the improvement in results with 35% replacement and the deterioration for higher percentages of replacement, as mentioned in the section corresponding to the compression strength test.

These tensile strength results are similar to those obtained in other studies. The results obtained by Aldahdooh [39] showed an improvement in the tensile and flexural strength of UHPFRC when 50% of the cement was replaced by palm oil fuel ash. Aldahdooh attributed this improvement to the higher content of amorphous $SiO_2$ present in this residue, which favors the pozzolanic reaction. I. Lopez et al. [14] observed the feasibility of replacing 100% of the silica fume with granite cutting waste without significant impact on the tensile strength of UHPC. These authors also attributed these good results to the irregular shape of the granite particles.

## 4. Conclusions

In view of the results obtained, it can be concluded that the use of granite cutting waste from gang saws (GCW-GS) is a viable alternative as partial substitutes for micronized quartz in the manufacture of ultra-high performance fiber reinforced concrete (UHPFRC).

Regarding the mechanical properties, an improvement can be appreciated in most of them. Therefore, the compressive strength increased for all the substitution percentages, up to a 14% for a 70% replacement, and the flexural strength and tensile strength increased up to a 12% for a 35% substitution. However, the modulus of elasticity of UHPFRC decreased up to a 8% for the 100% replacement. On the other hand, a loss of workability of fresh concrete was observed, which increased the demand of water.

Based on the tests carried out, it can be stated that in general, the GCW-GS waste improves the results by an average of approximately 5% over the results of GCW-D (which were available in previous studies), except for the modulus of elasticity, where there was a loss of 5% as compared to the GCW-D waste.

The final conclusion is that GCW-GS can be used for the manufacture of UHPFRC with an improvement of all the properties up to a 35% replacement. From this percentage, the results are at values close to the control concrete with no significant decreases, i.e., less than 10%, even for a 100% replacement.

**Author Contributions:** Conceptualization, F.L.G.; methodology, F.L.G. and I.L.B.; validation, F.L.G.; investigation, J.S.G. and C.L-C.P.; resources, J.S.G. and I.L.B.; writing—original draft preparation, I.L.B and J.S.G.; writing—review and editing, M.S.L. and J.S.G.; visualization, C.L.-C.P. and M.S.L.; supervision, C.L.-C.P. and F.L.G.; project administration, F.L.G.; funding acquisition, F.L.G. and J.S.G. All authors have read and agreed to the published version of the manuscript.

**Funding:** This research was funded by the Spanish Ministry of Economy and Competitiveness through the research project grant number BIA2016-78460-C3-2-R.

**Institutional Review Board Statement:** Not applicable.

**Informed Consent Statement:** Not applicable.

**Data Availability Statement:** Data presented are original and not inappropriately selected, manipulated, enhanced or fabricated.

**Acknowledgments:** The authors also want to thank the support of the following in carrying out this study: ArcelorMittal, Elkem, Basf, Sika AG, Grupo Minersa, Granitos Cabaleiro SA, and the Ministry of Economy and Competitiveness of the Government of Spain.

**Conflicts of Interest:** The authors declare no conflict of interest.

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
