# Peer review of "Use of Waste from Granite Gang Saws to Manufacture Ultra-High Performance Concrete Reinforced with Steel Fibers"

_applsci, doi:10.3390/app11041764_

Round 1

Reviewer 1 Report

The current study investigates the possibility of using waste collected from granite gang saws to produce new concrete with ultra high performance characteristics made from streel fibre. The reason for this is because the cutting process leaves a lot of useful waste that can be reused for useful applications and making concrete with special properties. The paper also compares the waste collected from two types of saw made from gang and diamond materials to find what are the contributions of calcium oxides and iron. The results were analysed by testing the compressive strength, elasticity and flexural and tensile properties. The results show that both waste collected from the two materials is useful for making the concrete with enhanced properties. The waste collected from gang saws showed better results due to presence of iron dioxide.

The abstract must be combined in one paragraph.

Combine the first two paragraphs in one, please avoid using small paragraphs which breaks the coherence of the paper

Line 42 please avoid bulk citations unless they are given full credit and detailed discussion of they have done and what were their main findings.

Please combine the introduction sections into bigger paragraphs, max three paragraphs for the whole introduction section as this way it makes the sequence broken and difficult to track the story in the paper

Figure 1 please add some arrows and text to tell the reader what they are looking at in these two images

Table 1 and 2 needs a reference if this was not data you measured or generated on your own.

Combine figures 2 and 3 and make them smaller as they are in the materials and method section so not to confuse the readers thinking they are results.

Table 4, what was the rationale behind these mixing values, is it based on industrial application or just to check a range of mixing levels to test the fabricated concrete samples?

Again please combine smaller figures into bigger ones as it is really difficult to follow the story this way.

Line 178-191 combine all this in one paragraph

Table 5 should have a +/- range since this is average, please add this data to the table

Line 195-204 please combine in one paragraph.

Figure 5 needs error bars to show range of the data

Figure 5 please add a more descriptive caption

Line 217-219 needs a reference to support this claim

Figure 8 could it be because of presence of voids, gaps or porosities in the mixture which caused reduction in the E modulus? Please explain this if possible.

Please use more descriptive captions for all figures, currently they are very short and do not provide sufficient description to what we are looking at in the figures.

Line 280-281 “….the loss of the strength” please support this claim with a reference

Did the authors measure the hardness of the fabricated samples, this might correlate well with module E and other properties studied here?

Figure 10 is missing caption!

The paper needs tidying and more analytical discussion on all observed results, please use past literature to support or compare your results with.

Conclusion too long and must be reduced, make it concise and to the point.

Author Response

All the changes in the manuscript are in red colour

REVIEWER 1

Comments and Suggestions for Authors

The abstract must be combined in one paragraph.

Combine the first two paragraphs in one, please avoid using small paragraphs which breaks the coherence of the paper

Thanks for your comment. The abstract has been rewritten in a shorter version in a single paragraph.

Line 42 please avoid bulk citations unless they are given full credit and detailed discussion of they have done and what were their main findings.

Thank you for the observation. This paragraph has been rewritten in a different way, so that a more detailed explanation about each citation is given in the following paragraphs.

Please combine the introduction sections into bigger paragraphs, max three paragraphs for the whole introduction section as this way it makes the sequence broken and difficult to track the story in the paper

Thanks for your suggestion. We have reduced the number of paragraphs to better track the development of the text.

Figure 1 please add some arrows and text to tell the reader what they are looking at in these two images

Thanks for your suggestion. A new figure has been added with an explanation of the image for better understanding.

Table 1 and 2 needs a reference if this was not data you measured or generated on your own.

Thank you for the comment. We have reviewed both tables and now we can state that all the results have been obtained on our own.

Combine figures 2 and 3 and make them smaller as they are in the materials and method section so not to confuse the readers thinking they are results.

Thanks for your suggestion. We have combined the two figures in an only one and made it a bit smaller.

Table 4, what was the rationale behind these mixing values, is it based on industrial application or just to check a range of mixing levels to test the fabricated concrete samples?

Effectively, it is based on an industrial application. Habitually, this mix is used in a precast concrete plant which we are working the last four years.

Again please combine smaller figures into bigger ones as it is really difficult to follow the story this way.

Thanks for your comment. We have combined images in figures 2 and 3.

Line 178-191 combine all this in one paragraph

Thanks for your comment. The whole text has been combined in a single paragraph.

Table 5 should have a +/- range since this is average, please add this data to the table

Thanks for your indication. We have added in the table the range of the obtained values. There is no range for the slump since an only test was performed for each mix.

Line 195-204 please combine in one paragraph.

Thanks for your comment. The text has been combined in a single paragraph.

Figure 5 needs error bars to show range of the data

Thank you for the observation. The mentioned figure refers to the values obtained for the slump. With regards to this test we have made one measurement for each mix, and that is the reason why we have not added error bars

Figure 5 please add a more descriptive caption

Thanks for your observation. A more descriptive caption has been added.

Line 217-219 needs a reference to support this claim

Thank you for the comment. We have added a reference to support the comment.

Figure 8 could it be because of presence of voids, gaps or porosities in the mixture which caused reduction in the E modulus? Please explain this if possible.

Yes, the elastic modulus reduction can be related with the porosity of the specimens but this reduction also can be occasioned by the a more weakness if the interface transition zone between the cement paste and the aggregates.

Please use more descriptive captions for all figures, currently they are very short and do not provide sufficient description to what we are looking at in the figures.

Thanks for the comment. The figure captions have been revised and better explained.

Line 280-281 “….the loss of the strength” please support this claim with a reference

Thank you for the observation. Two references have been added to support the statement.

Did the authors measure the hardness of the fabricated samples, this might correlate well with module E and other properties studied here?

No, we did not make test with the sclerometer. In any case this type of measures of the superficial hardness of the concrete can not be correlated with the elastic modulus.

Figure 10 is missing caption!

Thanks for the comment. An appropriate caption has been added.

The paper needs tidying and more analytical discussion on all observed results, please use past literature to support or compare your results with.

Thank you very much for your observation. We have improved the paragraph of analysis of results including several comparisons with the results obtained for other authors when discussing the results for each test.

Conclusion too long and must be reduced, make it concise and to the point.

Thank you for your comment. The conclusions section has been rewritten in a shorter and more concise way.

Reviewer 2 Report

Dear Authors,

Thank you for your well-written manuscript, here will be following comments:

Lines 13-27: your abstract have to be rewritten in a comprehensive paragraph, small story? (now reading it, it looks like a selection of key sentences) with focus on what is so novel in your study.

Lines 46-48: indeed it is possible to replace with high % levels but what about other properties – durability in that case? Is that reasonable? Please also mention other properties not only mechanical ones when you refer to high substitution levels.

Lines 48-55: repetition, can be shorten!

Line 224: Figure 5: and where is data of control mix?  In general you need to compare all mix modifications to reference one, please update you manuscript where is required! The same for Figure 6; In regards to formatting: please use values on y-axis (Fig.5  from 15 to 30, Fig. 6 2350 to 2550, etc.)

In regards to your described results – can you please compare those to other results by other authors and underline what is new in your paper? In general, it looks like description of basic experimental program/protocol. Please also move part of your conclusions into Discussion part and make shorter version of your conclusion with highlights of the current study.

Please make sure you follow the required formatting of your references.

Author Response

All the changes in the manuscript are in red colour

REVIEWER 2

Dear Authors,

Thank you for your well-written manuscript, here will be following comments:

Lines 13-27: your abstract have to be rewritten in a comprehensive paragraph, small story? (now reading it, it looks like a selection of key sentences) with focus on what is so novel in your study.

Thanks for your comment. We have rewritten the abstract shortening it in an only paragraph and underlining the novelty of the study.

Lines 46-48: indeed it is possible to replace with high % levels but what about other properties – durability in that case? Is that reasonable? Please also mention other properties not only mechanical ones when you refer to high substitution levels.

Thank you very much for your observation. We have explained better the consequences of high levels of substitution with regards to durability.

Lines 48-55: repetition, can be shorten!

Thanks for your appreciation. We have shortened the paragraph.

Line 224: Figure 5: and where is data of control mix?  In general, you need to compare all mix modifications to reference one, please update you manuscript where is required! The same for Figure 6; In regards to formatting: please use values on y-axis (Fig.5 from 15 to 30, Fig. 6 2350 to 2550, etc.)

Thanks for the comment. We have modified all the graphics according to your recommendation. With regard to control mix, it was present in the graphic with letter C, but for better clarity we have used the word “Control”.

In regards to your described results – can you please compare those to other results by other authors and underline what is new in your paper? In general, it looks like description of basic experimental program/protocol. Please also move part of your conclusions into Discussion part and make shorter version of your conclusion with highlights of the current study.

Thank you very much for your suggestion. We have improved the analysis of results including several references to the works carried out for other author and contrasting them with those obtained in or study. With regards to the conclusions paragraph, we have rewritten it in a more concise version.

Please make sure you follow the required formatting of your references

Thanks for your comment. We have reviewed the references.

Reviewer 3 Report

There are some weaknesses through the manuscript which need improvement. Therefore, the submitted manuscript cannot be accepted for publication in this form, but it has a chance of acceptance after a major revision. My comments and suggestions are as follows:

1- Abstract gives information on the main feature of the performed study, but some details about the materials and experimental tests (at least in a couple of sentences) should be added. However, a concise abstract is needed.

2- It is not necessary to present abstract in two separate paragraphs.

3- Authors must clarify necessity of the performed research (in introduction). The group citation (e.g., [2-7]) must be avoided. At least some of the reviewed reference must be comment in a couple of sentences.

4- The literature study must be enriched. It is highly recommended to read and cite the published papers: (a) https://doi.org/10.1080/19386362.2018.1505310 and (b) https://doi.org/10.1016/j.engstruct.2019.109844 Also, the main references of international standards must be cited.

5- Since the manuscript presents an experimental investigation, it is necessary to add figures to show experimental conditions. For instance, specimens under test (compression).

6- More details on experimental test must be presented. For example, presenting just results of compression is not enough (details must be presented in 3.3 and also details of other tests).

7-Authors must describe how elastic modulus of different mixtures are determined. Explanation and illustrating figures for subsections in section 3 are a necessity (showing just curves is not enough). For instance, details of tensile test and its figure must be added.

8- Please add figures, describe experimental tests, and present your manuscript in a scientific way.

9- In its language layer, the manuscript should be considered for English language editing. There are sentences which have to be rewritten.

10- The conclusion must be more than just a summary of the manuscript and bullets are not necessary. Please provide all changes in text and reference update (based on recommended papers) by red color in the revised version.

Author Response

All the changes in the manuscript are in red colour

REVIEWER 3

There are some weaknesses through the manuscript which need improvement. Therefore, the submitted manuscript cannot be accepted for publication in this form, but it has a chance of acceptance after a major revision. My comments and suggestions are as follows:

1- Abstract gives information on the main feature of the performed study, but some details about the materials and experimental tests (at least in a couple of sentences) should be added. However, a concise abstract is needed.

Thanks for your comment. The abstract has been rewritten in a shorter version and explained all the tests performed, as well as clarified the characteristics of the waste.

2- It is not necessary to present abstract in two separate paragraphs.

Thank you for the observation. The paragraph has been written in an only paragraph.

3- Authors must clarify necessity of the performed research (in introduction). The group citation (e.g., [2-7]) must be avoided. At least some of the reviewed reference must be comment in a couple of sentences.

Thank you for the observation. This paragraph has been rewritten in a different way, so that a more detailed explanation about each citation is given in the following paragraphs.

4- The literature study must be enriched. It is highly recommended to read and cite the published papers: (a) https://doi.org/10.1080/19386362.2018.1505310 and (b) https://doi.org/10.1016/j.engstruct.2019.109844 Also, the main references of international standards must be cited.

Thank for your suggestion. Both references have been added to the text. Regarding the standards that have been followed to perform the tests, although we referred to Spanish UNE Standards, all of them have an equivalent one in the European Standard (EN) with the same number. That is the reason why we have cited the standards as “UNE-EN”. Nevertheless, following your recommendation we have changed the references to EN.

5- Since the manuscript presents an experimental investigation, it is necessary to add figures to show experimental conditions. For instance, specimens under test (compression).

Thank you very much for your comment. Several figures about the process of making samples, as well as for different laboratory tests have been added.

6- More details on experimental test must be presented. For example, presenting just results of compression is not enough (details must be presented in 3.3 and also details of other tests).

Thanks for your comment. We have added more information about the results, including in table 6 the range of the values that we have obtained, and we have explained in detail the procedure to determine the tensile strength. With respect to the performed tests we have faithfully followed the indications of the international standards referred in the text. In paragraph 2.3.1 we have provided this information, as well as the number of test performed in each case.

7-Authors must describe how elastic modulus of different mixtures are determined. Explanation and illustrating figures for subsections in section 3 are a necessity (showing just curves is not enough). For instance, details of tensile test and its figure must be added.

Thanks for your comment. The modulus of elasticity was determined following in detail the international standard UNE-EN 12390-13. This method is based on applying to the specimen three compression load cycles, registering the curve stress-strength to determine then the slope of the secant line. With respect to the tensile strength test, we have added a detailed explanation of the procedure followed to determine indirectly this value including an explanatory figure for clarity.

8- Please add figures, describe experimental tests, and present your manuscript in a scientific way.

Thanks for your comment. We have added some images related to the tests performed to carry out the study: tensile strength and modulus of elasticity tests. As for the description of performed experimental tests, we have dispensed with them, since the explanations are described in detail in the mentioned international standards (EN).

9- In its language layer, the manuscript should be considered for English language editing. There are sentences which have to be rewritten.

Thank for your observation. The manuscript has been reviewed by a native English to correct all the possible mistakes.

10- The conclusion must be more than just a summary of the manuscript and bullets are not necessary.

Thank you for your comment. The conclusions section has been rewritten in a different way, removing the bullets and explaining the overall findings of the whole work.

Please provide all changes in text and reference update (based on recommended papers) by red color in the revised version.

Round 2

Reviewer 1 Report

authors have answered all questions

Reviewer 2 Report

Dear Authors, well done!

Please pay attention to line 182 (minor typo): "¿flexotraction? ¿flexural strength?"

Reviewer 3 Report

In the submitted revised manuscript, most of the reviewers' comments have been properly responded and corresponding modifications have been conducted. I think it can be considered for publication. (Please note that there are question marks in the text and Fig. 10 should be illustrated in a higher quality)